# What is the significance of a new breast mass in women aged 40 or above? A cross-sectional study

Sadaf Alipour[1,2], Kasra Jafari[3], Azin Saberi[2], Mandana Motamedi[1], Amirhossein Eskandari[4]*

1 Breast Diseases Research Center, Cancer Institute, Tehran University of Medical Sciences, Tehran, Iran, 2 Department of Surgery, Arash Women's Hospital, School of Medicine, Tehran University of Medical Sciences, Tehran, Iran, 3 Research Development Center, Arash Women Hospital, Tehran University of Medical Sciences, Tehran, Iran, 4 Deputy of Education, Ministry of Health, Tehran, Iran

* dr_a_eskandari@yahoo.com

## Abstract

### Background

Breast lumps are the most common presentation of breast cancer, and when detected in imaging, they are assigned a BI-RADS score dependent on their characteristics. Recency of a mass is not among these features, while in our center, the new occurrence of a solid breast mass in women 40 or above is considered suspicious because common benign masses typically occur before this age. We designed this study to evaluate data from our Arash Breast Diseases Registry to explore the results of this personalized approach.

### Methods

Women aged ≥40 with a breast mass in exam, ultrasound, or mammography that had been biopsied or followed-up for at least 2 years were included. In addition to BIRADS 4 and 5 or suspicious clinical exam, new solid masses on breast examination, mammography, or ultrasound in women 40 or above were biopsied. We calculated the sensitivity, specificity, likelihood ratios, and predictive values for our institutional protocol and the guideline-based protocol where the recency of the mass is not an indication for biopsy.

### Results

Among 4865 women, 609 were eligible. The mean age was 49.06 ± 7.84 years; 135 were malignant, and 474 were benign. Our protocol showed an increase in the sensitivity (84.1% vs. 70.1%) and a decrease in specificity (40.3% vs. 95.4%) when compared to the usual protocol.

**Data availability statement:** The data that support the findings of this article are not publicly available due to ethical concerns observed in the instructions of the Arash Registry of Breast Diseases Protocol. They can be requested from bdrc@tums.ac.ir.

**Funding:** This study was supported by a grant from the Deputy of Research of Tehran University of Medical Sciences, Tehran, Iran; Grant Number 1400-1-259-53034. The funders had no role in study design, data collection and analysis, decision to publish, or preparation of the manuscript.

**Competing interests:** The authors have declared that no competing interests exist.

**Abbreviations:** NCCN, National Comprehensive Cancer Network; MG, mammography; US, ultrasound; MRI, magnetic resonance imaging; ACR, American College of Radiology; BI-RADS, Breast Imaging Reporting and Data System; BC, Breast cancer; TUMS, Tehran University of Medical Sciences; BE, Breast exam; PLR, positive likelihood ratio; NLR, negative likelihood ratio; PPV, positive predictive value; NPV, negative predictive value; OC, ovarian cancer.

## Conclusions

While our institutional protocol increased the sensitivity and could lead to early detection, it lowered specificity, suggesting more false positive results. Our main suggestion is to consider "recency of a solid mass" as a criterion for suspiciousness in BI-RADS assignment to breast masses. We propose further studies to achieve a protocol that maximizes both sensitivity and specificity without compromising diagnostic reliability, and to find the cut-off for age when the effect of this factor increases.

## Introduction

Breast symptoms are one of the most common complaints of women, and one of the most important findings in the breast is lumps. Although more commonly benign in nature [1], they are the most frequent clinical presentation of breast cancer (BC) [2]. The approach to breast lumps follows some international guidelines in many medical centers. For example, the National Comprehensive Cancer Network (NCCN) is an international guideline that states a planned approach to breast symptoms, including masses, according to the patient's age, classifying it as those detected above or below 30 years of age. However, the algorithm only considers palpable masses [3]. Alternatively, national or local principles are observed in some other centers for the selection of management options regarding breast masses.

On the other hand, masses detected in mammography (MG), breast ultrasound (US), or breast magnetic resonance imaging (MRI) are defined in the American College of Radiology (ACR) Breast Imaging Reporting & Data System (BI-RADS). The assignment of a BI-RADS score to a mass is dependent on the visualized characteristics of that mass and the associated findings. The specific assigned BI-RADS, generally from B2 to B5, determines the management strategy that the radiologist would propose [4].

The possibility for a mass to be malignant increases with age and with a positive family history [1,5,6]. Although logically, these items should be considered while approaching a finding in the breast, these items are not included in the BI-RADS lexicon. In our institute, one other feature is also regarded as suspicious, and this one is not specified in BI-RADS or existing guidelines. For us, the new occurrence of a mass in the breast per se is regarded as suspicious. The reason for this is that common benign masses, such as fibroadenomas, typically occur before the age of forty, and other benign lesions are less likely to present as a mass [7]. Thus, in our center, the approach to a solid breast mass that has recently appeared in a woman 40 years of age or above is similar to suspicious masses, and biopsy is strongly considered for most of them.

In Iran, most private offices and public clinics dedicated to breast diseases have set their principles according to the priority of answering cancer cases, and a few accept and visit women who come for opportunistic BC screening. Also, most

specialized and sub-specialized breast centers only accept women with a BI-RADS 4 or 5 imaging, or with a histological diagnosis of BC. The Breast Clinic of Arash Women Hospital is one of the highest-flow outpatient referral centers for breast diseases in the country and admits women who attend for general breast complaints or for their regular or random breast checkup and screening. Data of all women attending our Breast Clinic of Arash Women's Hospital is being entered in the "Arash Breast Diseases Registry" supported by the Registry office of Tehran University of Medical Sciences, which constitutes the only one of its kind in Iran. We wanted to explore the results of our personalized approach to patients 40 or above with a new solid breast mass regarding BC diagnosis, so we designed the present study to evaluate data from two subsequent years completed in the Registry.

## Methods

### Study design

This cross-sectional study has been approved by the Ethics Committee of Tehran University of Medical Sciences (IR. TUMS.SINAHOSPITAL.REC.1400.028) and uses the data of the Arash Breast Diseases Registry (Approval code: 99-2-422-46300, Ethics code: IR.TUMS.DDRI.REC.1399.036). Consent to participate requirements was waived by the Institutional Ethics Committee as the data were gathered retrospectively from medical records.

The study population consisted of all women who had attended the Breast Clinic in Arash Women Hospital, Tehran, Iran, from April 2014 to March 2016 for opportunistic BC screening or for breast complaints. Data of women aged 40 and above who were under surveillance for at least one year prior to and in whom a breast mass had been found in breast exam (BE), breast ultrasound (US), or mammography (MG) were extracted. Those who had less than 2 years of follow-up, while a biopsy of the mass had not been done, were excluded.

The benign or malignant nature of the masses was considered based on the histology results in those who had undergone biopsy, and according to the exam and imaging findings, and the course of the lesion during the follow-up in those who had not been sampled. The imaging and BE findings, the personal and family cancer history of the patients, and their reproductive features were extracted from the Registry at 2024-06-01.

### Diagnosis protocols

Our institutional protocol that is carried out in the Breast Clinic for women who attend for opportunistic screening includes a BE done by a Breast Surgeon, Oncologic Surgeon, or General Surgeon dedicated to breast diseases; an MG for women 40 years of age or above, a US for those who have dense breasts on mammogram (ACR 3 or 4) or who need a complementary assessment based on the MG and BE results, extra mammogram views or MRI as recommended by the radiologist, and biopsy when needed. In the hospital, the imaging is performed or interpreted by radiologists dedicated to the breast, but if women come with imaging results from known centers, these are considered appropriate, and the imaging is not repeated unless dubious. In addition, many centers outside the hospital refer their patients for image findings. These can include suspicious findings as well as BI-RADS 0 results. The decision to biopsy is dependent on the results of each of the three modalities, including the BE, the US, and the MG. A new mass detected in these assessments in women 40 or above is an indication for biopsy unless it is shown to be a simple or complicated cyst, a typical lipoma, or an intramammary lymph node at US. The definition of a new mass in this protocol includes finding a mass in an imaging modality that had not been detected on previous imaging within one year, or a mass found by the patient or by physical breast exam that was absent in a previous exam. If a patient had undergone a breast exam or imaging for the first time, the mass was considered new. Suspicious clinical breast exam consists of findings in favor of malignancy, like a hard, irregular, fixed, dominant, or new mass; new nipple or skin retraction, pathologic nipple discharge, or focal asymmetric areas of thickening or firmness. Fig 1 illustrates our institutional protocol for these cases in brief.

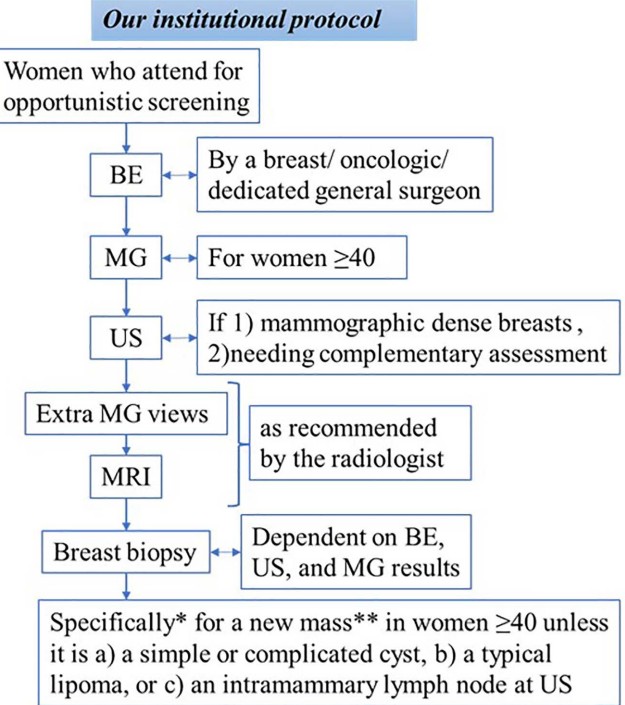

**Fig 1. Flowchart of our institutional protocol for women who attend for opportunistic screening.** *Regarding the present research, **A new mass = a mass absent on prior imaging or BE within 1 year. BE = Breast exam, MG = Mammography, US = Breast ultrasound.

## Sample size

The sample size was calculated to estimate the sensitivity of the diagnostic assessment for detecting breast cancer among patients referred because of a new breast mass. We planned to detect a sensitivity (Se) of 95% with a two-sided 95% confidence level and a marginal error (d) of 0.05. The required number of diseased participants was computed using the following formula:

$$n_{Se\ (Unadjusted\ for\ prevalence)} = \frac{Z^2_{\frac{\alpha}{2}} \times Se \times (1 - Se)}{d^2}$$

$$n_{Se\ (Unadjusted\ for\ prevalence)} = \frac{1.96^2 \times 0.95 \times (1 - 0.95)}{0.05^2} \cong 73$$

Based on our prior data from our center, approximately 20% of referred cases with a new breast mass have breast cancer (prevalence = 0.20); therefore, the total sample size required to have at least 73 diseased participants is:

$$n_{Se\ (Adjusted\ for\ prevalence)} = \frac{n_{Se}}{Prevalence}$$

$$n_{Se\ (Adjusted\ for\ prevalence)} = \frac{73}{0.2} = 365$$

Thus, a total of 365 participants were required to ensure approximately 73 cancer cases and to estimate a sensitivity = 95% with ±5% marginal error at the 95% confidence level.

## Statistical analysis

We calculated the sensitivity, specificity, positive likelihood ratio (PLR), negative likelihood ratio (NLR), positive predictive value (PPV), and negative predictive value (NPV) for our institutional protocol, and then for a presumed situation where we would have followed existing guidelines (the guideline-based protocol), it means without considering the recency of the mass as an indication for biopsy. For these estimations, we defined the gold standard as the histology results in those who had been biopsied, and the result of the 2-year follow-up in those who had not.

T-tests (or Mann-Whitney) were performed to compare continuous parametric and non-parametric variables, respectively. The $\chi 2$ test (or Fisher's Exact) was used to analyze ratios. P-values $< 0.05$ were used as the cutoff point for significance. We used Stata version 17.0 for statistical analysis.

## Results

Among 4865 women who had visited the Breast Clinic in the study period, 2686 were 40 years old or above, and 609 of these had a new breast mass. The mean age of the patients at the time of the first detection of the mass was $49.06 \pm 7.84$ (range, 40–86) years. The mean follow-up time was $64.29 \pm 22.85$ months.

Out of 609 cases, 135 were BC, and 474 were benign. Patients with malignant new masses were significantly older, more at the menopausal stage of their life, and reported a self-history of BC or ovarian cancer (OC) and all types of cancers (p-value $< 0.05$) more frequently. Also, gravidity was significantly different between the two groups, while the highest numbers of gravidity ($\geq 5$) were more commonly associated with malignant cases (34.81%) compared to benign ones (14.56%), and lower gravidity (0 and 1) had a slightly higher prevalence in benign cases. The overall characteristics of all the participants according to the malignant or benign nature of their mass are demonstrated in Table 1.

Table 2 shows the mammographic breast density in all the patients, and the MG and US BI-RADS of the benign and malignant masses.

Fig 2 shows the course of the diagnostic process in benign and malignant masses according to the BI-RADS at imaging modalities and the cause for further assessment despite a low MG BI-RADS score.

The results of comparing the diagnostic values of both protocols are presented in Table 3. Our institutional protocol shows an increase in the sensitivity (84.1% compared to 70.1%) but a substantial decrease in the specificity (40.3% compared to 95.4%) when compared to the guideline-based protocol. None of the tumors that were recognized as benign and did not undergo biopsy, as well as those that were biopsied and yielded a benign result, transformed to malignant in the follow-up period.

## Discussion

This study was carried out to evaluate the results of our specific approach to breast masses in women 40 and above, where recency of the mass is considered a suspicious feature regardless of the other characteristics. We found a few BCs that would not have been diagnosed at that time without this approach.

Many of the disorders and diseases of the breast might present as masses, including the most common breast diseases such as fibroadenomas or breast cysts [8], much less frequent disorders like granulomatous mastitis and hamartomas [9], and malignancies of the breast. While benign lumps are important due to disturbing symptoms, their dominant concern consists of the risk of malignant transformation [10]; meanwhile, the first and foremost concern of physicians is the accurate early differentiation from an existing cancer.

In a study by Lumachi et al. [5] in 2002, the evaluation of around 2879 patients with breast complaints in a 14-year experience was reported, and 318 patients had BC. Similar to our study, benign versus malignant diagnosis had been performed by biopsy or at least a two-year follow-up. They classified patients according to age and type of complaints. They showed that the risk of BC increased with age and was higher when the presentation consisted of a breast lump.

**Table 1. Characteristics of all participants with a new mass (n = 609).**

| Characteristics | | Benign (n = 474) | Malignant (n = 135) | P-value |
|---|---|---|---|---|
| Age | Mean ± SD | 47.77 ± 6.50 | 53.96 ± 10.16 | **<0.001** |
| | Range | 40-80 | 40-86 | |
| Menarche age | Mean ± SD | 13.34 ± 1.57 | 13.33 ± 1.39 | 0.963 |
| | Range | 9-19 | 10-17 | |
| Gravidity | 0 | 35 (7.38) | 7 (5.19) | **<0.001** |
| | 1 | 35 (7.38) | 10 (7.41) | |
| | 2 | 150 (31.65) | 28 (20.74) | |
| | 3 | 112 (23.63) | 27 (20) | |
| | 4 | 73 (15.4) | 16 (11.85) | |
| | ≥ 5 | 69 (14.56) | 47 (34.81) | |
| Menopausal status* | Premenopausal | 337 (71.1) | 57 (42.22) | **<0.001** |
| | Menopause | 137 (28.9) | 78 (57.78) | |
| Hx of HRT* | No | 461 (99.78) | 129 (99.23) | 0.337 |
| | Yes | 1 (0.22) | 1 (0.77) | |
| Hx of OCP use* | No | 374 (81.3) | 95 (74.8) | 0.106 |
| | Yes | 86 (18.7) | 32 (25.2) | |
| Infertility* | No | 440 (96.28) | 125 (97.66) | 0.449 |
| | Yes | 17 (3.72) | 3 (2.34) | |
| Self-Hx of BC or OC* | No | 464 (98.51) | 109 (81.95) | **<0.001** |
| | Yes | 7 (1.49) | 24 (18.05) | |
| Self-Hx of All Cancers* | No | 462 (98.09) | 109 (81.95) | **<0.001** |
| | Yes | 9 (1.91) | 24 (18.05) | |
| FH** of BC or OC* | No | 400 (84.57) | 117 (86.67) | 0.546 |
| | Yes | 73 (15.43) | 18 (13.33) | |

* Number (Percent), ** In first- or second-degree family members. BC = breast cancer, OC = ovarian cancer, FH = family history, HRT = hormone replacement therapy, Hx = history, OCP = oral contraceptive pills, SD= standard deviation.

In our study, results showed that women with benign tumors were younger, had fewer pregnancies, were more frequently premenopausal, and were less likely to have a history of previous cancer (Table 1, p < 0.001 for all these variables). These are in line with known factors of risk, as BC occurrence is directly associated with age [11]. The premenopausal status and the lower rate of pregnancy might be due to the younger age of the benign group, but parity is per se a well-known (weak) preventive factor against BC [11–13].

In this study, according to our institutional protocol, the reason for the US was a dense MG in 143 patients, and around 160 patients underwent US because of a BI-RADS 0 MG. After ruling out cysts, all new solid masses in women 40 and above were biopsied, and nearly half were malignant. Other studies have shown lower rates of malignancy in biopsied solid breast masses. In 2007, Seltzer [14] gave a detailed report of the outcome of investigations he conducted for 10000 patients who attended his surgery clinic for breast problems. All of them either underwent biopsy or were followed till benignity or malignancy was ascertained. Of 10000 patients, less than half had breast lumps, less than half of the lumps underwent biopsy, and about one-fourth of the biopsied masses were cancers. The percentage of patients with a breast mass that was finally diagnosed as BC was 47% in those aged 70 or more, 19% in 50–69 years, 6% 30–49 years, and 1% in those less than 30 years. This study did not assess the recency of the mass, and the rates of malignancy were much higher in our study. This implies that the newness of the mass was the factor that induced this increment, and it can be considered a weighty BC risk factor. However, in our study, of those that were biopsied only because of the recency

 

**Table 2. Mammographic breast density and BI-RADS of mammography and breast ultrasound in benign and malignant masses.**

| Imaging Results | | Benign (n = 474) Number (%) | Malignant (n = 135) Number (%) | P-value |
|---|---|---|---|---|
| Mammography BI-RADS | B0 | 151 (36.47) | 17 (15.89) | **<0.001** |
| | B1 | 97 (23.43) | 4 (3.74) | |
| | B2 | 121 (29.23) | 6 (5.61) | |
| | B3 | 26 (6.28) | 5 (4.67) | |
| | B4_any | 9 (2.17) | 30 (28.04) | |
| | B4a | 5 (1.21) | 1 (0.93) | |
| | B4b | 5 (1.21) | 3 (2.8) | |
| | B4c | 0 (0) | 7 (6.54) | |
| | B5 | 0 (0) | 34 (31.78) | |
| Sonography BI-RADS | B0 | 1 (0.26) | 1 (1) | **<0.001** |
| | B1 | 13 (3.32) | 1 (1) | |
| | B2 | 199 (50.77) | 9 (9) | |
| | B3 | 110 (28.06) | 5 (5) | |
| | B4 | 69 (17.61) | 42 (42) | |
| | B4_any | 29 (7.4) | 23 (23) | |
| | B4a | 33 (8.42) | 7 (7) | |
| | B4b | 6 (1.53) | 4 (4) | |
| | B4c | 1 (0.26) | 8 (8) | |
| | B5 | 0 (0) | 42 (42) | |

BI-RADS = Breast Imaging Reporting & Data System.

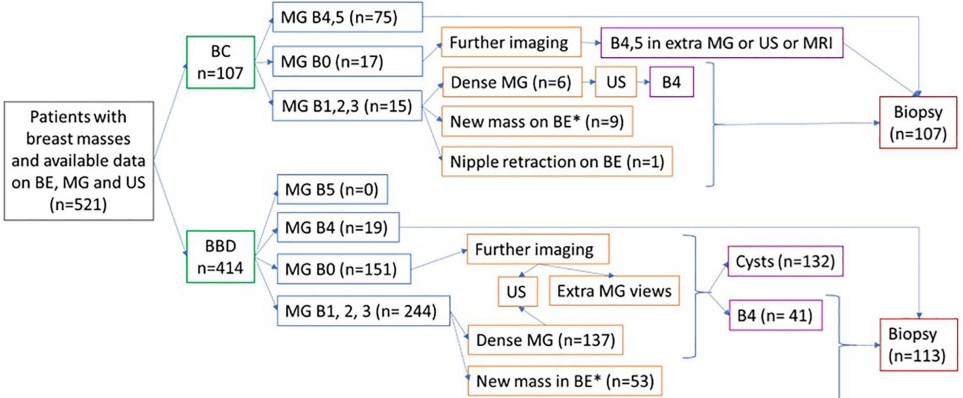

**Fig 2. Diagnostic course of benign and malignant masses; the benign masses that were not biopsied were proven as such by their benign course during the 2-year follow-up.** *Not cystic in ultrasound. B = BI-RADS, BBD = benign breast disease, BC = breast cancer, BE = breast exam, MG = mammography, US = ultrasound.

factor and had otherwise no reason for it, around 16% were malignant. This is a low rate, and it can be guessed that these would have enlarged and gotten biopsied in the next 6–12 months, but the effect of that delay on prognosis might be considerable. On the other hand, the 84% rate of negative biopsy is high. The advantage of early diagnosis versus the high rate of negative biopsies should be balanced against each other in further prospective studies, but for now, consideration of the recency of a mass as a significant factor in decision-making shows sound reasoning.

**Table 3. Diagnostic values of the protocol of our center and the usual protocol.**

|  | Sensitivity (95% CI) | Specificity (95% CI) | ROC area (95% CI) | PLR (95% CI) | NLR (95% CI) | PPV (95% CI) | NPV (95% CI) |
|---|---|---|---|---|---|---|---|
| Guideline-based protocol | 70.1% (60.5–78.6) | 95.4% (92.9–97.2) | 0.83 (0.78–0.87) | 15.27 (9.68–24.10) | 0.31 (0.23–0.42) | 79.8% (70.2–87.4) | 92.5% (89.5–94.8) |
| Institutional protocol* | 84.1% (75.8–90.5) | 40.3% (35.6–45.2) | 0.62 (0.58–0.66) | 1.41 (1.26–1.58) | 0.39 (0.25–0.62) | 26.7% (22.1–31.8) | 90.8% (85.6–94.5) |

* The protocol in our center, where all new breast masses in women aged 40 or above are biopsied. CI = confidence interval, NLR = negative likelihood ratio, NPV = negative predictive value, PLR = positive likelihood ratio, PPV = positive predictive value.

BI-RADS is a tool that provides a robust structure for a thorough assessment of breast images and a straightforward conclusion about the risk of the findings and the subsequent steps to take [4]. Imaging criteria have been defined to esti-mate the nature of a mass and recommend an approach accordingly. For example, an irregular shape and an indistinct margin are in favor of malignancy, and a higher BI-RADS is assigned to such a mass. Accordingly, a regular mass will be assigned a BI-RADS 2 or 3, whether new or not, and recommended to be followed up in 6 or 12 months. Globally, nurses and midwives, general practitioners, or physicians in various disciplines who have not been specifically trained in breast management are in charge of visiting women with breast complaints; also, in many countries, BC screening is done by MG, and no BE is performed. Therefore, the end recommendation of the imaging report is the basis for decision-making about breast findings, and the false negative rate (FNR) should be minimized. The FNR is up to 1.5 per 1000 women in digital MG BC screening [15], and 0.8–5.4% in diagnostic digital MG [16]. The FNR of breast US is around 6%. These FNRs might decrease if other factors are considered in the BI-RADS classification.

Overall, some characteristics are extremely important in the clinical approach to a breast mass. For example, the probability for a mass to be malignant increases with age [17,18]. A previous history of BC affects the approach to a mass found during the follow-up of BC survivors. In healthy people, finding a breast mass in the background of a positive family history of breast or ovarian cancer, especially in males or young females of the family, strongly moves our diagnostic steps toward more precise ones, such as biopsy or MRI. Ackerman et al. [19] performed a study on 1201 patients to assess the accuracy of BI-RADS in the US of solid breast masses with specific attention to BI-RADS 3 lesions. They showed that considering the clinical features and history of patients and masses decreases the false negative rate, and conclude that while adherence to BI-RADS is necessary, these factors should also be considered in deciding for biopsy of solid breast masses.

Our institutional protocol demonstrates a notable increase in sensitivity (84.1% vs. 70.1%), indicating a higher true positive rate. This suggests that our institutional protocol is more effective in identifying patients with breast malignancies. However, this increased sensitivity comes at the cost of a significant reduction in specificity (40.3% vs. 95.4%), leading to a higher false positive rate. Consequently, the PPV drops substantially from 79.8% to 26.7%, implying that a positive test result is less reliable in confirming the presence of malignancy under our institutional protocol. The NPV remains relatively high for both protocols (92.5% for the guideline-based protocol and 90.8% our institutional protocol), ensuring that a nega-tive result is still a strong indicator that malignancy is not present.

The ROC area, which combines sensitivity and specificity, decreases from 0.83 to 0.62 in our institutional protocol, reflecting a reduction in overall diagnostic accuracy. Likelihood ratios, which are instrumental in assessing the diagnostic test's performance, show a drastic decrease in our institutional protocol (PLR from 15.27 to 1.41 and NLR from 0.31 to 0.39). This indicates that our institutional protocol is less effective in altering the pre-test probability of malignancy.

While the new protocol increases sensitivity and could potentially lead to early detection, its lower specificity and PPV suggest that it may result in more false positives, possibly leading to unnecessary invasive procedures. Actually, the main drawback of our proposed (institutional) protocol is the high rate of biopsy. Thus, the balance between sensitivity

and specificity must be carefully considered to optimize patient outcomes and avoid undue harm. Further studies and refinements may be necessary to achieve a protocol that maximizes both sensitivity and specificity without compromising diagnostic reliability.

The rate of biopsy appears high in this study. One reason is the referral nature of our center, and the fact that the population under study was not under an organized BC screening program. On the other hand, the policy of considering new masses for tissue diagnosis increases the rate of biopsy. This point is the main drawback of our protocol.

To our knowledge, no study has investigated the significance of the recency of a breast mass in BC diagnosis. We believe that one valuable feature that should lower the threshold for biopsy is the development of a new mass in women 40 and above.

Our study had some limitations. First, the sample size was not large enough. Second, we chose the age of 40 because it is the age of BC screening, and also because the mean age of BC is around 46 years in Iran, younger than in Western countries [20]. We have not assessed the difference in other age limits. Finally, since the dataset does not identify the specific radiologists for each assignment, calculating inter-rater reliability is not possible.

## Conclusion

We showed that "recency of the mass" is an important factor in the assessment of solid breast masses at 40 years of age or above. Our main suggestion is to consider it in approaching breast lumps and include it as an effective criterion in the BI-RADS lexicon. We propose that similar studies, including a higher number of masses, be carried out to compare the effect of including or not including the recency factor in clinical decision-making about breast masses, and to find the cut-off for age when the effect of this factor increases.

## Acknowledgments

We would like to acknowledge Miss Sarah Ahmadizadeh, Mrs. Matina Noori, and Mrs. Maryam Vahab for their assistance in data gathering for this project. Also, we acknowledge the Deputy of Research of TUMS for financially supporting this project, and the Registry Office of TUMS for supporting the Arash Breast Diseases Registry.

During the preparation of this work, the authors used ChatGPT (developed by OpenAI, 2025) in order to improve the readability and language of this paper. After using this tool, the authors reviewed and edited the content as needed and take full responsibility for the content of the published article.

## Author contributions

**Conceptualization:** Sadaf Alipour, Kasra Jafari, Azin Saberi, Mandana Motamedi, Amirhossein Eskandari.

**Data curation:** Kasra Jafari.

**Formal analysis:** Kasra Jafari.

**Funding acquisition:** Sadaf Alipour, Amirhossein Eskandari.

**Investigation:** Sadaf Alipour, Azin Saberi, Mandana Motamedi, Amirhossein Eskandari.

**Methodology:** Sadaf Alipour, Kasra Jafari, Azin Saberi, Mandana Motamedi, Amirhossein Eskandari.

**Project administration:** Sadaf Alipour, Amirhossein Eskandari.

**Supervision:** Sadaf Alipour, Amirhossein Eskandari.

**Validation:** Amirhossein Eskandari.

**Writing – original draft:** Sadaf Alipour, Kasra Jafari, Azin Saberi, Mandana Motamedi, Amirhossein Eskandari.

**Writing – review & editing:** Sadaf Alipour, Kasra Jafari, Azin Saberi, Mandana Motamedi, Amirhossein Eskandari.

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
