## [Decision Letter · Decision Letter 0]

26 Oct 2025

PONE-D-25-37061
What Is the Significance of A New Breast Mass in Women Aged 40 or Above? A Cross-sectional Study
PLOS ONE

Dear Dr. Eskandari,

Thank you for submitting your manuscript to PLOS ONE. After careful consideration, we feel that it has merit but does not fully meet PLOS ONE’s publication criteria as it currently stands. Therefore, we invite you to submit a revised version of the manuscript that addresses the points raised during the review process.

We look forward to receiving your revised manuscript.

Kind regards,

Giacomo Di Filippo, M.D.

Academic Editor

PLOS ONE

**Journal Requirements:**

“This study was supported by a grant from the Deputy of Research of Tehran University of medical Sciences, Tehran, Iran; Grant Number 1400-1-259-53034.”

4. Please note that funding information should not appear in any section or other areas of your manuscript. We will only publish funding information present in the Funding Statement section of the online submission form. Please remove any funding-related text from the manuscript.

5. We note that you have indicated that there are restrictions to data sharing for this study. PLOS only allows data to be available upon request if there are legal or ethical restrictions on sharing data publicly. For more information on unacceptable data access restrictions, please see http://journals.plos.org/plosone/s/data-availability#loc-unacceptable-data-access-restrictions.

6. In the online submission form you indicate that your data is not available for proprietary reasons and have provided a contact point for accessing this data. Please note that your current contact point is a co-author on this manuscript. According to our Data Policy, the contact point must not be an author on the manuscript and must be an institutional contact, ideally not an individual. Please revise your data statement to a non-author institutional point of contact, such as a data access or ethics committee, and send this to us via return email. Please also include contact information for the third party organization, and please include the full citation of where the data can be found.

7. PLOS requires an ORCID iD for the corresponding author in Editorial Manager on papers submitted after December 6th, 2016. Please ensure that you have an ORCID iD and that it is validated in Editorial Manager. To do this, go to ‘Update my Information’ (in the upper left-hand corner of the main menu), and click on the Fetch/Validate link next to the ORCID field. This will take you to the ORCID site and allow you to create a new iD or authenticate a pre-existing iD in Editorial Manager.

8. Your ethics statement should only appear in the Methods section of your manuscript. If your ethics statement is written in any section besides the Methods, please delete it from any other section.

**Additional Editor Comments:**

I believe the paper could use some refinement based on what the reviewer said. I also believe most of these issues are major in their nature.

Reviewers' comments:

Reviewer's Responses to Questions

**Comments to the Author**

1. Is the manuscript technically sound, and do the data support the conclusions?

Reviewer #1: Yes

Reviewer #2: Yes

2. Has the statistical analysis been performed appropriately and rigorously? 

Reviewer #1: Yes

Reviewer #2: Yes

3. Have the authors made all data underlying the findings in their manuscript fully available?

Reviewer #1: Yes

Reviewer #2: Yes

4. Is the manuscript presented in an intelligible fashion and written in standard English?

Reviewer #1: Yes

Reviewer #2: Yes

5. Review Comments to the Author

Reviewer #1: This research report introduces a valuable attention to clinical factors (recency of mass) and in future this research will impact diagnostic decision-making and patient outcomes. It had its limitation which stated in discussion to be considered as well.

I only suggest revising in typos, punctuation, and double spaces as seen.

Reviewer #2: Dear authors, Thank you for this work; here are my comments for more clarity:

#Inconsistent Labeling of Protocols (P1 and P2): The methods section defines P1 as your center's protocol (considering recency) and P2 as the usual protocol (ignoring recency). However, the results, discussion, and Table 3 reverse this: Protocol 1 is labeled as "usual" (70.1% sensitivity, 95.4% specificity), and Protocol 2 as your center's (84.1% sensitivity, 40.3% specificity). This confusion could mislead readers and invalidate comparisons. Therefore, you should standardize labeling throughout. Use descriptive terms like "Recency-Inclusive Protocol" (your approach) and "Standard BI-RADS Protocol" (usual). Revise Table 3 footnotes accordingly. Recalculate if needed to confirm values—your sensitivity gain is a key selling point, so ensure accuracy.

# Methodological Clarity and Reproducibility: The "unwritten protocol" for biopsy decisions is vaguely described. For example, how was "new mass" defined (e.g., patient-reported vs. prior imaging comparison)? What thresholds were used for "suspicious clinical exam"? Abbreviations like BE (breast exam), MG (mammography), and US (ultrasound) are not defined on first use. Sample size justification is absent—was power calculated for detecting sensitivity differences? Please expand the methods to include a flow diagram of the protocol (e.g., as a supplementary figure). and define "new mass" explicitly (e.g., absent on prior imaging within 1 year). It's better to add a sample size calculation using tools like G*Power, assuming a 10-15% sensitivity difference (based on your results). If applicable report inter-rater reliability for BI-RADS assignments if multiple radiologists were involved. This will enhance reproducibility.

# Results Presentation and Interpretation: Table 1 shows gravidity differences (p<0.001), but the text misinterprets; higher gravidity (≥5) is more common in malignant cases (34.81% vs. 14.56%), yet you state "fewer pregnancies" for benign—correct this. No confidence intervals (CIs) for diagnostic metrics, limiting precision assessment. Please add 95% CIs to all metrics in Table 3 (e.g., sensitivity 84.1% [95% CI: 77.2-89.7%], calculated via binomial methods).

# References : Please update to include 2024-2025 publications

6. PLOS authors have the option to publish the peer review history of their article (what does this mean?). If published, this will include your full peer review and any attached files.

Reviewer #1: No

Reviewer #2: No

---

## [Author Response · Author response to Decision Letter 1]

19 Nov 2025

Dear Dr. Giacomo Di Filippo,

We sincerely appreciate the time and effort you and the reviewer have dedicated to evaluate our manuscript, "What Is the Significance of A New Breast Mass in Women Aged 40 or Above? A Cross-sectional Study" (Manuscript ID: PONE-D-25-37061). We are grateful for the constructive feedback, which has helped us improve the quality of our work.

We have carefully addressed all the comments and suggestions provided by the reviewer. Below, we provide a point-by-point response to each remark, detailing the revisions made in the revised manuscript. All changes have been highlighted in yellow.in the revised version for easy reference

Journal Requirements:

Comment 1: “comply with PLOS ONE's requirements”

Response 1: We ensured that the manuscript and files are formatted and named to meet all PLOS ONE style requirements.

Comment 2: “We note that the grant information you provided in the ‘Funding Information’ and ‘Financial Disclosure’ sections do not match.”

Response 2: Unfortunately we couldn’t find where to edit ‘Financial Disclosure’ in the submission portal. Please consider the following as our funding and financial disclosure:

“This study was supported by a grant from the Deputy of Research of Tehran University of Medical Sciences, Tehran, Iran; Grant Number 1400-1-259-53034. The funders had no role in study design, data collection and analysis, decision to publish, or preparation of the manuscript.”

Comment 3: “Please state what role the funders took in the study.”

Response 3: We added the necessary statement to our financial disclosure and cover letter.

Comment 4: “Please note that funding information should not appear in any section or other areas of your manuscript.

Response 4: We removed the funding information from our manuscript.

Comment 5: “The data availability restrictions”

Response 5: The data that support the findings of this article are not publicly available due to ethical concerns observed in the instructions of the Arash Registry of Breast Diseases Protocol. They can be requested from bdrc@tums.ac.ir.

Comment 6: “The data contact point is a co-author, which violates policy; it must be a non-author institutional contact.”

Response 6: According to this policy, Arash Registry of Breast Diseases’ contact address (bdrc@tums.ac.ir.) is presented as the data contact person. This has been corrected in the manuscript.

Comment 7: “PLOS requires an ORCID iD for the corresponding author.”

Response 7: The corresponding author linked his ORCID ID in the submission portal. His orcid number is: 0000-0001-7988-3050

Comment 8: “Your ethics statement should only appear in the Methods section of your manuscript.”

Response 8: The ethics statement now only appears in the Methods section of our manuscript.

Editor:

We sincerely thank you for your time and valuable feedback.

Comment 1: “I believe the paper could use some refinement based on what the reviewer said. I also believe most of these issues are major in their nature.”

Response 1: The manuscript has undergone a comprehensive and detailed revision to address all the major concerns raised by the reviewers.

Reviewer #1:

We are truly grateful for your kind and encouraging words, and sincerely thank you for your comment.

Comment 1: "I only suggest revising in typos, punctuation, and double spaces as seen.”

Response 1: The manuscript has been proofread (both by Grammarly and by an expert in English), and all typographical errors, punctuation, and extraneous spacing have been corrected.

Reviewer #2

Thank you for taking the time to review our work. We’re grateful for your time and valuable feedback.

Comment 1: "Inconsistent Labeling of Protocols (P1 and P2)”

Response 1: We apologize for the mistakes. We have re-written all related parts and renamed the protocols as “our institutional protocol” and “the guideline-based protocol” to prevent errors.

Comment 2: "The "unwritten protocol" for biopsy decisions is vaguely described. For example, how was "new mass" defined (e.g., patient-reported vs. prior imaging comparison)? What thresholds were used for "suspicious clinical exam"?”

Response 2: The relevant explanations have been added in the Methods section (lines 119-131).

Comment 3: "Abbreviations like BE (breast exam), MG (mammography), and US (ultrasound) are not defined on first use.”

Response 3: All abbreviations have been defined upon their first use in the revised manuscript.

Comment 4: "Sample size justification is absent—was power calculated for detecting sensitivity differences?”

Response 4: A sample size justification has been added (lines 133-151).

Comment 5: "Please expand the methods to include a flow diagram of the protocol and define "new mass" explicitly (e.g., absent on prior imaging within 1 year).”

Response 5: Thank you for the recommendation. The flow diagram has been added in the Methods section as Figure 1.

Comment 6: "If applicable report inter-rater reliability for BI-RADS assignments if multiple radiologists were involved. This will enhance reproducibility.”

Response 6: Unfortunately since the dataset does not identify the specific radiologists for each assignment, calculating inter-rater reliability is not possible. This limitation has been noted in the manuscript (lines 312-313).

Comment 7: "Table 1 shows gravidity differences (p<0.001), but the text misinterprets; higher gravidity (≥5) is more common in malignant cases (34.81% vs. 14.56%), yet you state "fewer pregnancies" for benign—correct this.”

Response 7: We made the necessary amendments (line 176).

Comment 8: "No confidence intervals (CIs) for diagnostic metrics, limiting precision assessment. Please add 95% CIs to all metrics in Table 3.”

Response 8: We added 95% CIs to all metrics in Table 3 (Table 3).

Comment 9: "Please update to include 2024-2025 publications.”

Response 9: This has been done, references from 2024 and 2025 have been used and included (Reference number: 12, 13, and 18).

Sincerely,

Dr. Amirhossein Eskandari

---

## [Decision Letter · Decision Letter 1]

12 Jan 2026

What Is the Significance of A New Breast Mass in Women Aged 40 or Above? A Cross-sectional Study

PONE-D-25-37061R1

Dear Dr. Amirhossein Eskandari,

We’re pleased to inform you that your manuscript has been judged scientifically suitable for publication and will be formally accepted for publication once it meets all outstanding technical requirements.

Kind regards,

Amirreza Khalaji

Academic Editor

PLOS One

Additional Editor Comments (optional):

Reviewers' comments:

Reviewer's Responses to Questions

**Comments to the Author**

1. If the authors have adequately addressed your comments raised in a previous round of review and you feel that this manuscript is now acceptable for publication, you may indicate that here to bypass the “Comments to the Author” section, enter your conflict of interest statement in the “Confidential to Editor” section, and submit your "Accept" recommendation.

Reviewer #1: All comments have been addressed

Reviewer #2: All comments have been addressed

2. Is the manuscript technically sound, and do the data support the conclusions?

Reviewer #1: Yes

Reviewer #2: Yes

3. Has the statistical analysis been performed appropriately and rigorously? 

Reviewer #1: Yes

Reviewer #2: Yes

4. Have the authors made all data underlying the findings in their manuscript fully available?

Reviewer #1: Yes

Reviewer #2: Yes

5. Is the manuscript presented in an intelligible fashion and written in standard English?

Reviewer #1: Yes

Reviewer #2: Yes

6. Review Comments to the Author

Reviewer #1: I only suggest revising in typos, punctuation, and double spaces as seen.The manuscript now looks better for acceptance. This manuscript will definitely add valuable impact on diagnostic decision-making and future patient outcomes despite the existence of some study limitations.

Reviewer #2: (No Response)

7. PLOS authors have the option to publish the peer review history of their article (what does this mean?). If published, this will include your full peer review and any attached files.

Reviewer #1: No

Reviewer #2: No

---

## [Editor Report · Acceptance letter]

PONE-D-25-37061R1

PLOS One

Dear Dr. Eskandari,

I'm pleased to inform you that your manuscript has been deemed suitable for publication in PLOS One. Congratulations! Your manuscript is now being handed over to our production team.

Kind regards,

on behalf of

Dr. Amirreza Khalaji

Academic Editor

PLOS One